# Associations of sunlight affinity with depression and sleep disorders in American males: Evidence from NHANES 2009–2020

Haifeng Liu[1], Jia Yang[1]*, Tiejun Liu[2], Weimin Zhao[3]

1 College of Chinese Medicine, Changchun University of Chinese Medicine, Changchun City, Jilin Province, China, 2 Department of Gastroenterology, First Affiliated Hospital to Changchun University of Chinese Medicine, Changchun City, Jilin Province, China, 3 Department of Neurology, First Affiliated Hospital to Changchun University of Chinese Medicine, Changchun City, Jilin Province, China

☯ These authors contributed equally to this work.

* Yangjia202416@163.com

## Abstract

### Objective

Depression and sleep disorders are globally prevalent, yet male-specific studies remain scarce. This study investigates associations between sunlight affinity (a novel dual-dimensional metric comprising psychological [sunlight preference score, SPS] and behavioral [sunlight exposure duration, SED] dimensions) and subthreshold depression (StD), major depressive disorder (MDD), short sleep, and trouble sleeping in American males.

### Methods

We analyzed weighted data from 7,306 males in the National Health and Nutrition Examination Survey (2009–2020) and assessed sunlight affinity's associations with depression and sleep disorders based on multiple logistic regression, threshold effects analysis, restricted cubic spline (RCS) analysis, subgroup analysis, and mediation analysis.

### Results

Adjusted multiple logistic regression analyses showed SPS inversely associated with StD (OR = 0.88, 95% confidence interval [CI]: 0.80–0.96) and MDD (OR = 0.80, 95% CI: 0.69–0.92), but positively with short sleep (OR = 1.11, 95% CI: 1.04–1.19). SED negatively correlated with MDD (OR = 0.90, 95% CI: 0.84–0.96) and trouble sleeping (OR = 0.94, 95% CI: 0.90–0.98), while positively with short sleep (OR = 1.05, 95% CI: 1.01–1.10). The highest SED quartile had reduced StD risk (OR = 0.70, 95% CI: 0.52–0.94). RCS analysis revealed a U-shaped relationship between SPS and short sleep (P-nonlinearity = 0.003). Threshold analyses identified SPS inflection

**Data availability statement:** The data file for this study has been deposited in the figshare data repository and is available at the following link: https://doi.org/10.6084/m9.figshare.30173047. This dataset can be freely downloaded by anyone.

**Funding:** The author(s) received no specific funding for this work.

**Competing interests:** The authors have declared that no competing interests exist.

points: ≥ 2.867 linked to higher short sleep risk (OR=1.17, 95% CI: 1.08–1.26) and ≥4 to lower trouble sleeping (OR=0.62, 95% CI: 0.48–0.80). Subgroup analyses revealed significant interactions across different populations. Mediation analysis suggested potential suppression effect of sunlight affinity in the bidirectional cycles between depression and sleep disorders.

## Conclusion

This study revealed that sunlight affinity was inversely associated with depression and trouble sleeping and positively associated with short sleep in males. Further longitudinal studies are needed to confirm causality.

---

## Introduction

Depression and sleep disorders are significant public health issues, particularly among males, who face unique challenges in diagnosis and treatment [1]. The global burden of depressive disorders affects 280 million people [2], where males manifest significant clinical severity, including a 3–4 fold greater suicide mortality rate compared to females, however, the underdiagnosis of depression in males, attributed to the insidious nature of depressive symptoms and negative healthcare-seeking attitudes, results in a diagnosis rate approximately half that of females [3,4], posing significant challenges to effective treatment. Subthreshold depression (StD), recognized as a prodromal stage of major depressive disorder (MDD), has a prevalence roughly three times greater than that of MDD among the general populace, with about 10–20% of StD cases progressing to MDD [5,6]. Consequently, proactive prevention strategies aiming to block StD to MDD transition and halt MDD aggravation are essential for population health management, especially in males. It is noteworthy that, sleep disorders are emerging as a growing public mental health concern, particularly manifesting as insufficient sleep duration in male populations [7]. Moreover, conditions like obstructive sleep apnea and sleep fragmentation synergistically worsen male sleep health. Furthermore, underdiagnosis intensifies the burden of sleep disorders in males [8]. More importantly, depression and sleep disorders often co-occur, with approximately 90% of individuals diagnosed with MDD experiencing sleep disturbances [2]. This bidirectional relationship underscores the need for integrated approaches to treatment.

Although recent advances in light therapy have garnered significant attention for its applications in depression and sleep disorders [9], sunlight exposure remains under-explored despite its potential as a cost-effective and adjustable factor for mental health. A Study indicated that each incremental hour of sunlight exposure correlates with a 4% decrease in insomnia incidence, 19% reduction in fatigue severity, and 4% attenuation of lifetime major depressive disorder risk [10]. Sunlight alleviates sleep disorders by multiple potential mechanisms: modulating circadian rhythms, regulating melatonin secretion, and enhancing vitamin D synthesis [11,12]. Importantly, psychological attitudes toward sunlight possibly play a critical role. A Google search-based

analysis demonstrated a strong inverse relationship between sunlight exposure and searches for depressive suicidal language, suggesting that sunlight may foster optimistic perspectives [13]. However, compared with the behavioral aspects of sunlight exposure, the psychological dimensions, particularly attitudes toward sunlight, remain underexplored and underrecognized in clinical research. Previous research found that males exhibit stronger preferences for sunlight and greater physiological sensitivity. A global cross-sectional study involving 50,552 participants revealed that the proportion of males actively seeking sunlight exposure within the past 12 months was as high as 85.04% [14]. Additionally, males show heightened circadian and emotional sensitivity to variations in light [15]. Nevertheless, the relationship of sunlight between depression and sleep disorders in males remains unclear.

By leveraging the National Health and Nutrition Examination Survey (NHANES) database and employing various statistical methodologies, our study addresses the limitations of conventional research that predominantly focuses on population-level commonalities by specifically targeting the United States male individuals. The biophilia theory proposes that people have an innate tendency to connect with the natural world. This natural affinity is expressed as "nature con-nectedness." Research shows that being connected to nature is associated with pro-environmental behaviors, better health, and greater happiness [16]. Building on this framework, we propose the novel metric of "sunlight affinity." This inte-grated metric combines psychological dimensions (sunlight preference score, SPS) and behavioral dimensions (sunlight exposure duration, SED). The two dimensions could be integrated through synergistic neurobehavioral pathways. Neu-robiological studies show intrinsically photosensitive retinal ganglion cells (ipRGCs)–nucleus accumbens circuits encode light as rewarding (SPS), driving approach behaviors (SED) via dopamine [17]. Prefrontal mechanisms link behavioral exposure (SED) to enhanced psychological preference (SPS) may be through cognitive learning [18]. Physiologically, SED enhances mood through vitamin D synthesis and circadian entrainment [19]. The sunlight affinity was used to com-prehensively elucidate the relationships between sunlight exposure and both depression and sleep disorders in males. The findings could provide novel insights for assessing depression and sleep disorders among American males.

## Materials and methods

### Research design

This was a cross-sectional study. We have made use of de-identified public data from NHANES, the administration of which is the responsibility of the Centers for Disease Control and Prevention, with data access complying with their policies (https://wwwn.cdc.gov/nchs/nhanes/). A sophisticated multi-stage probability sampling design is employed by NHANES, incorporating home-based interviews, screenings, and laboratory tests conducted at mobile examination centers. The study procedures received approval from the National Center for Health Statistics Research Ethics Review Board, and informed written consent was obtained from participants before data collection began (https://www.cdc.gov/nchs/nhanes/about/erb.html). Given the use of aggregated, anonymized data with no individual identifiers, additional ethi-cal clearance was not needed.

### Study population

This investigation analyzed resources from NHANES 2009–2020 (initial sample size, n = 55,999). This period was selected for its comprehensive coverage of sunlight affinity indicators (SPS and SED), the Patient Health Questionnaire-9 (PHQ-9) scores, and the sleep disorder measures, all of which are critical for examining health behaviors in males. NHANES employs a multistage probability sampling methodology developed by the Centers for Disease Control and Prevention to produce health information that is representative at the national level. Participants aged <20 years (n = 23,501) were excluded to avoid physiological and behavioral instability during adolescence. Given that NHANES categorizes par-ticipants solely based on biological sex and considering the specific focus of our study population, female participants (n = 16,768) were excluded from the analysis. Additionally, those with missing key variables, including sunlight affinity indicators (SPS, n = 5,453; SED, n = 946; total n = 6,399), PHQ-9 scores (n = 1,218), sleep disorder measures (n = 26), or

incomplete covariates (e.g., demographics, lifestyle factors, comorbidities; n = 781), were excluded. The NHANES questionnaire modules regarding sunlight preference and exposure duration were administered only to participants aged 20–59 years. Consequently, individuals aged 60 years or older were excluded from this analysis to ensure the valid assessment of the primary exposure variable, sunlight affinity. The final cohort comprised 7,306 United States adult males with complete data. The participant selection procedure is illustrated in Fig 1.

## Assessment of sunlight affinity

Sunlight affinity is a dual-dimensional metric combining the SPS and SED to measure individuals' psychological and behavioral proximity to sunlight. The SPS was assessed through the question: "When you go outside on a very sunny day for more than one hour, how often do you stay in the shade?" Responses were rated on a 0–5 scale: "don't go out in the sun" (0), "always" (1), "most of the time" (2), "sometimes" (3), "rarely" (4), and "never" (5). The higher the score, the

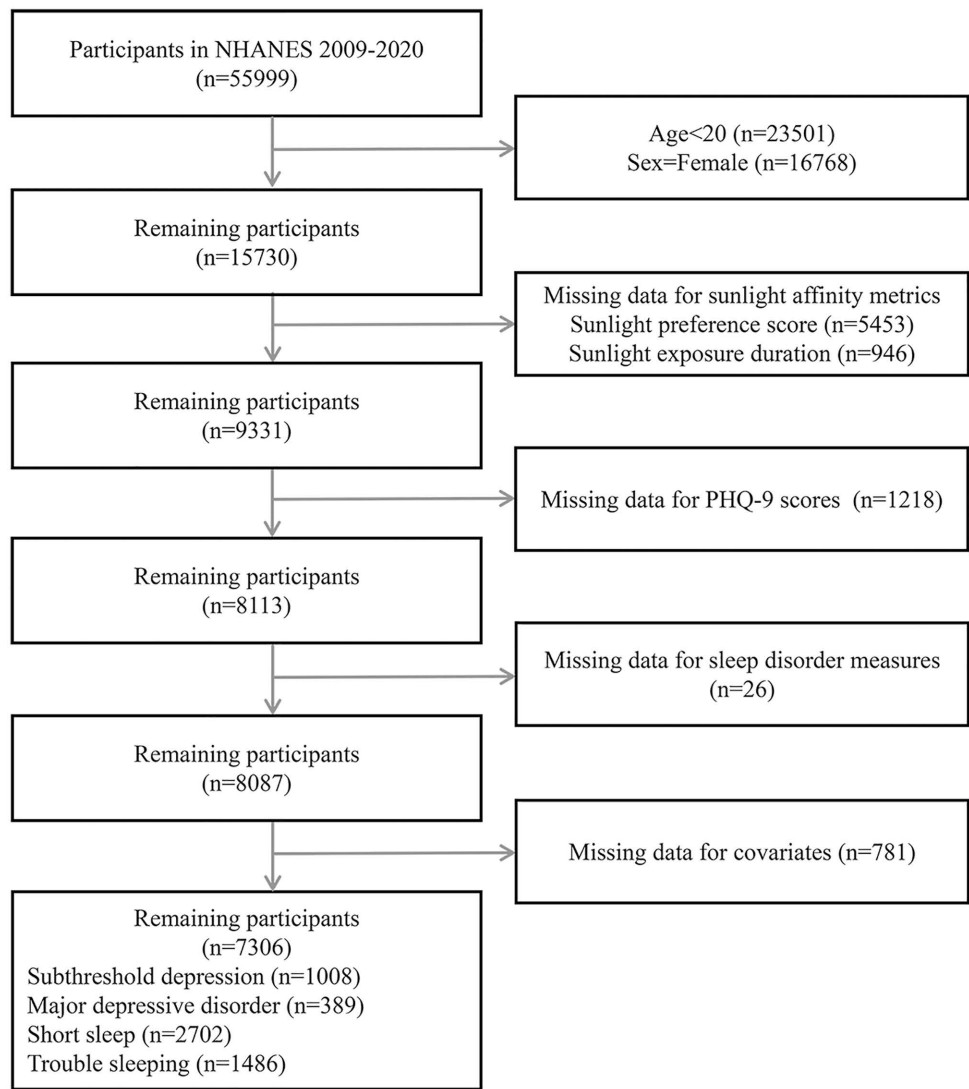

**Fig 1. Flow chart.**

greater the preference for sunlight. Based on their survey responses, participants were assigned to three groupings: negative attitude (SPS < 3), neutral attitude (SPS = 3), and positive attitude (SPS > 3).

The definition of SED was based on participants' reports of the time (hours) they spent outdoors (outside and not under any shade) between 9:00 am and 17:00 over the previous 30 days, including both working and non-working days. Mean values derived from participants' responses during working and non-working days served as a proxy measure for daily sunlight exposure time. This approach aligns with established methodologies in prior research [20]. However, constrained by the NHANES database, SED solely measures sunlight exposure duration and cannot incorporate variations in ultraviolet B radiation intensity influenced by geographical locations or seasons. SED was stratified into quartiles (Q1: ≤ 1 h, Q2: 1–2 h, Q3: 2–4 h, Q4: 4–8 h) to capture exposure gradients. Data was collected via standardized Computer-Assisted Personal Interviewing questionnaires, ensuring response consistency. Answers indicating "don't know" or refusal responses were coded as missing values and subsequently excluded from the analysis.

We observed a positive correlation (S1 Table) between SPS and SED (r = 0.184, P < 0.001). The variance inflation factors for both variables were 1.035, well below the threshold of 5, indicating no significant multicollinearity in the regression models. Although the correlation is relatively weak, its significance suggests a potential association between psychological preferences and behavioral exposure, which is consistent with the biophilia hypothesis. Simple linear regression analyses (S2 Table) further demonstrated a positive correlation between increases in the SPS score and an average increase of 0.35 hours in SED (β = 0.35, 95% confidence interval [CI] = 0.31–0.40, P < 0.001), while each hour increase in SED corresponded to an average increase of 0.10 points in SPS (β = 0.10, 95% CI = 0.08–0.11, P < 0.001). These preliminary findings support the coherence of the dual-dimensional metric.

## Assessment of depression and sleep disorders

The PHQ-9, a widely adopted tool for assessing depression severity, was designed by the American Psychiatric Association and is supported by solid psychometric validation [21]. In this study, the PHQ-9 was utilised to evaluate depression, and the instrument demonstrated both sensitivity and specificity of 88%. Participants rated their experiences using the following response options: "not at all" (0), "several days" (1), "more than half the days" (2), and "nearly every day" (3). The evaluation of each item is conducted on a scale ranging from 0 to 3, with the total score thus expressed as a number between 0 and 27 [22]. In this research, depression severity was classified as follows: scores of 0–4 represented no depression; 5–9 denoted StD [23]; and scores of 10 or above signified MDD [24].

Sleep disorders were measured with two indicators: "trouble sleeping" and "sleep hours." Assessment of trouble sleeping was based on the item: "Have you ever told a doctor or other health professional that you have trouble sleeping?" Individuals responding "yes" to the question were considered to have trouble sleeping and were therefore included in the study. Sleep hours were derived from self-reported sleep time. For participants in the 2009–2014 survey cycles, this was based on the direct question "How much sleep do you usually get at night on weekdays or workdays?" (NHANES variable SLD010H; answer in hours). For the 2015–2020 cycles, sleep duration was calculated as the time difference between the responses to " What time do you usually go to sleep on weekdays or workdays?" (SLQ300) and "What time do you usually wake up on weekdays or workdays?" (SLQ310) [25]. This computed value (in hours) provided a consistent measure of usual weekday sleep duration across all cycles. Individuals reporting sleep hours of fewer than 7 were categorized as having short sleep [26]. Answers indicating "don't know" or refusal responses were coded as missing values and subsequently excluded from the analysis.

## Covariates assessment

We examined factors influencing depression and sleep by incorporating various demographic, lifestyle, and comorbidity data into our statistical analysis. The demographic variables comprised age, race, education, marital status and poverty income ratio (PIR). Lifestyle variables encompassed smoking history (characterized by a history of smoking over 100

cigarettes during one's lifespan), alcohol consumption history (characterized by consumption of more than 12 alcoholic beverages within a 12-month period), physical activity (classification was based on Metabolic Equivalent of Task (MET) scores, low physical activity was designated as a MET/week value of less than 500, while high physical activity was defined as a value of 500 or more MET/week), and body mass index (BMI) [27]. Comorbidities included cardiovascular disease (CVD), liver condition, asthma, weak/failing kidneys, and cancer/malignancy, these comorbidities were assessed through self-reporting. A participant was recognized as having diabetes according to American Diabetes Association criteria if he or she had a self-reported diagnosis by a physician, or was taking medication to regulate blood glucose, or had an HbA1c ≥ 6. 5%, or an oral glucose tolerance test ≥200 mg/dL, or fasting plasma glucose ≥126 mg/dL. Hypertension was diagnosed if the average of three blood pressure readings exceeded 140/90 mmHg or by self-reported physician diagnosis [28].

## Statistical methodology

This study employed multiple statistical methods, weighted in line with National Center for Health Statistics guidelines to address the complex sampling design. The presentation of baseline characteristics was as the mean value ± standard error for continuous variables that were normally distributed. Comparisons of multiple groups were conducted using analysis of variance. In cases where the variable did not have a normal distribution, data were expressed as median (IQR, interquartile range), with subsequent group comparisons conducted utilizing the Kruskal-Wallis test. In order to assess the normality of the distribution, the Kolmogorov-Smirnov test was conducted. Categorical data were presented as a percentage and subjected to analysis using either the chi-squared test or the Fisher's exact test, depending on the nature of the data. To evaluate associations between sunlight affinity and depression/sleep disorders, three multivariable logistic regression models were formulated: The first model (Model 1) was without adjustment for covariates, whereas the second model (Model 2) only adjusted for demographic covariates (including age, race, education, marital status and PIR). Refer to S3 Table for details. The third model (Model 3) further adjusted for lifestyle (including smoking history, alcohol consumption history, physical activity, and BMI) and comorbidities (including CVD, liver condition, asthma, weak/failing kidneys, cancer/malignancy, diabetes, and hypertension) based on Model 2. The absence of substantial multicollinearity was confirmed in the fully adjusted model (Model 3), with all variance inflation factors being below 3. In Model 3, nonlinear relationships were evaluated using restricted cubic spline (RCS) regression, incorporating four optimally selected knots to achieve model flexibility-parsimony balance while mitigating overfitting risks [25]. Threshold effects were also assessed to identify potential dose-response patterns employing a two-piecewise linear regression model, whilst likelihood ratio tests were used to examine whether the two-piecewise model was comparable to a linear model. To evaluate the heterogeneity of the associations identified in Model 3, subgroup and interaction analyses for demographic covariates were conducted using multivariable logistic regression models. Nonparametric bootstrap mediation analyses (1,000 resamples; random seed = 123) were performed to evaluate the indirect effect (IE) of sunlight affinity across three pathways: Sleep hours and PHQ-9 scores, trouble sleeping and PHQ-9 scores, and sleep disorders (short sleep/trouble sleeping) and depression (StD/MDD). The IE was estimated from the product of the exposure-mediator coefficients and the mediator-outcome coefficients. Total effect (TE) was derived as the sum of IE and direct effect (DE: exposure–outcome association adjusted for the mediator). The proportion mediated was quantified as IE/TE × 100%. Participants with zero values for SPS or SED might have been unwilling to be exposed to sunlight due to potential health issues or might have been unable to go outdoors due to time constraints. To assess the sensitivity of the associations, we reconstructed the multivariable logistic regression, RCS, and threshold effect analysis after excluding these participants with zero SPS/SED values. The results were then compared with those of the original Model 3 to evaluate consistency. All analyses maintained the covariate adjustment strategy of Model 3. The goodness-of-fit for multivariable logistic regression models was assessed using the Akaike information criterion (AIC) and the Bayesian information criterion (BIC), with lower values indicating a better balance of model fit and parsimony.

The significance threshold for this study was established at a two-tailed P < 0.05. Data processing and analysis were conducted using R 4.3.2, along with Zstats v1.0 (www.zstats.net).

## Results

### Baseline characteristics of participants

This cross-sectional survey analyzed 7,306 United States males aged 20–59 years, including 1,397 participants with depression (StD: 1,008; MDD: 389) and 4,193 with sleep disorders (short sleep: 2,707; trouble sleeping: 1,486). The mean ages were 38 years for depression, 40 years for short sleep, and 43 years for trouble sleeping. Weighted analyses demonstrated significant differences in demographic, lifestyle, and comorbidity distributions (P < 0.05). Notably, depressed individuals had fewer positive attitudes toward sunlight compared to non-depressed individuals (P = 0.011), but the short sleep group had more positive attitudes toward sunlight compared to non-short sleep group (P < 0.001). Participants with the longest SED (Q4) exhibited higher proportions of the short sleep group compared to the non-short sleep group (P = 0.004), but lower proportions in trouble sleeping group compared to the non-trouble sleeping group (P = 0.004). Refer to Table 1 for details.

Non-normality of the continuous variables was confirmed by the Kolmogorov-Smirnov test (P < 0.05). Non-normally distributed continuous data were expressed as median ($Q_1$, $Q_3$), and group comparisons were performed using the Kruskal-Wallis test. Categorical data were presented as n (%) and analyzed using the chi-square test or Fisher's exact test, as appropriate. A P-value of less than 0.05 was considered statistically significant for all between-group comparisons.

### Associations of sunlight affinity with depression and sleep disorders

Our three logistic regression models demonstrated consistent directional associations. In the fully adjusted Model 3, we observed potential correlations between sunlight affinity and risks of depression and sleep disorders in males. For depression, each score gain in SPS was related to a 12% decrease in the odds of StD (aOR = 0.88, 95% CI: 0.80–0.96) and 20% decreased likelihood of MDD (aOR = 0.80, 95% CI: 0.69–0.92). By contrast to individuals with negative attitudes toward sunlight (SPS < 3), those with positive preferences (SPS > 3) exhibited 31% reduced probability of StD (aOR = 0.69, 95% CI: 0.53–0.92). A significant linear trend was observed (P-trend = 0.009). Notably, even a neutral attitude (SPS = 3) was connected to lower odds of MDD (aOR = 0.60, 95% CI: 0.39–0.92), and the linear trend was found to be statistically significant (P-trend = 0.005). RCS analysis indicated no significant nonlinear association between SPS and depression (P-nonlinear >0.05). Furthermore, each hour of SED was linked to 10% reduction in MDD odds (aOR = 0.90, 95% CI: 0.84–0.96). In comparison with the lowest SED quartile (Q1), the longest quartile (Q4) was related to 32% lower odds of MDD (aOR = 0.68, 95% CI: 0.48–0.96), and a significant linear trend emerged for SED and MDD (P-trend = 0.037); While the remaining quartiles showed inverse associations with StD (P < 0.05), with the Q4 demonstrating 30% decreased odds (aOR = 0.70, 95% CI: 0.52–0.94), and an L-shaped nonlinear association was revealed by RCS analysis between SED and StD (P-nonlinear = 0.018).

As for sleep disorders, each score increases in the SPS related to an 11% increase in the risk of short sleep (aOR = 1.11, 95% CI: 1.04–1.19). Individuals with a positive attitude toward sunlight (SPS > 3) were 34% more likely to experience short sleep (aOR = 1.34, 95% CI: 1.12–1.62) compared to those with a negative attitude toward sunlight (SPS < 3) and showed a significant linear trend (P-trend = 0.001). A U-shaped nonlinear relationship between SPS and short sleep was found by RCS analysis (P-nonlinear = 0.003), and threshold effect analysis showed that SPS ≥ 2.867 was significantly correlated with an increased probability of short sleep (aOR = 1.17, 95% CI: 1.08–1.26). Interestingly, threshold effect analysis suggested that SPS ≥ 4 was related to a 38% reduction in the likelihood of trouble sleeping (aOR = 0.62, 95% CI: 0.48–0.80). Moreover, each hour of SED corresponded to 5% elevated odds of short sleep (aOR = 1.05, 95% CI: 1.01–1.10) but 6% decreased likelihood of trouble sleeping (aOR = 0.94, 95% CI: 0.90–0.98). In comparison with the Q1, Q2 (the second quartile) showed a 20% decrease in the probability

**Table 1. Characteristics of participants by depression and sleep disorders in baseline.**

| Variable | Non-depression (n=5909) | StD (n=1008) | MDD (n=389) | P | Non-short sleep (n=4604) | Short sleep (n=2702) | P | Non-trouble sleeping (n=5820) | Trouble sleeping (n=1486) | P |
|---|---|---|---|---|---|---|---|---|---|---|
| **Age (year), M (IQR)** | 39.00 (20.00) | 38.00 (20.00) | 38.00 (21.00) | 0.659 | 38.00 (21.00) | 40.00 (18.00) | <0.001 | 38.00 (20.00) | 43.00 (19.00) | <0.001 |
| **PIR, M (IQR)** | 3.43 (3.25) | 2.65 (3.32) | 1.85 (3.17) | <0.001 | 3.31 (3.36) | 3.19 (3.37) | 0.150 | 3.14 (3.40) | 3.73 (3.22) | <0.001 |
| **BMI (kg/m², M (IQR)** | 28.00 (7.25) | 28.30 (7.6) | 28.00 (8.70) | 0.393 | 27.87 (7.30) | 28.50 (7.40) | 0.003 | 27.80 (7.20) | 29.20 (8.20) | <0.001 |
| **Race, n (%)** | | | | 0.586 | | | <0.001 | | | <0.001 |
| Mexican American | 917 (10.31) | 152 (10.68) | 47 (8.55) | | 728 (10.35) | 388 (10.14) | | 972 (11.70) | 144 (5.61) | |
| Non-Hispanic Black | 1220 (9.96) | 205 (9.98) | 71 (9.86) | | 779 (7.86) | 717 (13.96) | | 1232 (10.73) | 264 (7.43) | |
| Non-Hispanic White | 2238 (64.22) | 412 (64.48) | 179 (66.97) | | 1868 (66.30) | 961 (60.73) | | 2053 (61.08) | 776 (75.23) | |
| Other Hispanic | 560 (6.65) | 96 (7.13) | 47 (8.24) | | 440 (6.52) | 263 (7.31) | | 585 (7.44) | 118 (4.65) | |
| Other race | 974 (8.86) | 143 (7.73) | 45 (6.38) | | 789 (8.97) | 373 (7.86) | | 978 (9.05) | 184 (7.07) | |
| **Education, n (%)** | | | | <0.001 | | | 0.005 | | | <0.001 |
| <High school | 1048 (12.40) | 205 (14.08) | 111 (19.32) | | 873 (12.65) | 491 (13.57) | | 1161 (14.33) | 203 (8.51) | |
| High school | 1362 (23.23) | 269 (27.17) | 102 (27.92) | | 1042 (22.50) | 691 (26.82) | | 1400 (24.18) | 333 (23.37) | |
| > High school | 3499 (64.38) | 534 (58.75) | 176 (52.76) | | 2689 (64.85) | 1520 (59.61) | | 3259 (61.50) | 950 (68.12) | |
| **Marital status, n (%)** | | | | <0.001 | | | 0.792 | | | <0.001 |
| Never married | 1065 (17.25) | 218 (18.76) | 90 (21.36) | | 887 (17.89) | 486 (17.20) | | 1125 (18.13) | 248 (16.08) | |
| Widowed/Divorced/Separated | 937 (15.70) | 246 (26.01) | 113 (29.38) | | 784 (17.55) | 512 (18.18) | | 953 (16.31) | 343 (22.55) | |
| Married/Living with partner | 3907 (67.05) | 544 (55.23) | 186 (49.25) | | 2933 (64.56) | 1704 (64.62) | | 3742 (65.56) | 895 (61.37) | |
| **Smoke, n (%)** | | | | 0.569 | | | 0.771 | | | <0.001 |
| Yes | 1220 (23.20) | 211 (24.85) | 79 (21.09) | | 963 (23.47) | 547 (23.03) | | 1136 (21.37) | 374 (29.70) | |
| No | 4689 (76.80) | 797 (75.15) | 310 (78.91) | | 3641 (76.53) | 2155 (76.97) | | 4684 (78.63) | 1112 (70.30) | |
| **Alcohol, n (%)** | | | | 0.262 | | | 0.003 | | | 0.660 |
| Yes | 3735 (62.70) | 635 (62.01) | 236 (56.65) | | 2816 (60.75) | 1790 (65.29) | | 3682 (62.11) | 924 (62.97) | |
| No | 2174 (37.30) | 373 (37.99) | 153 (43.35) | | 1788 (39.25) | 912 (34.71) | | 2138 (37.89) | 562 (37.03) | |
| **Physical activity, n (%)** | | | | 0.230 | | | 0.416 | | | 0.008 |
| Low physical activity | 1281 (19.74) | 223 (22.17) | 111 (24.03) | | 1007 (19.94) | 608 (20.93) | | 1258 (19.37) | 357 (23.26) | |
| High physical activity | 4628 (80.26) | 785 (77.83) | 278 (75.97) | | 3597 (80.06) | 2094 (79.07) | | 4562 (80.63) | 1129 (76.74) | |
| **CVD, n (%)** | | | | <0.001 | | | 0.024 | | | <0.001 |
| Yes | 179 (2.89) | 54 (4.12) | 39 (8.83) | | 148 (2.91) | 124 (4.19) | | 165 (2.63) | 107 (5.70) | |
| No | 5730 (97.11) | 954 (95.88) | 350 (91.17) | | 4456 (97.09) | 2578 (95.81) | | 5655 (97.37) | 1379 (94.30) | |
| **Liver condition, n (%)** | | | | <0.001 | | | 0.905 | | | <0.001 |
| Yes | 179 (2.86) | 47 (3.50) | 32 (7.86) | | 153 (3.18) | 105 (3.23) | | 161 (2.39) | 97 (5.86) | |
| No | 5730 (97.14) | 961 (96.50) | 357 (92.14) | | 4451 (96.82) | 2597 (96.77) | | 5659 (97.61) | 1389 (94.14) | |
| **Asthma, n (%)** | | | | <0.001 | | | 0.241 | | | <0.001 |
| Yes | 737 (12.45) | 163 (15.20) | 89 (23.56) | | 591 (12.94) | 398 (14.18) | | 691 (11.75) | 298 (18.66) | |
| No | 5172 (87.55) | 845 (84.80) | 300 (76.44) | | 4013 (87.06) | 2304 (85.82) | | 5129 (88.25) | 1188 (81.34) | |
| **Diabetes, n (%)** | | | | <0.001 | | | 0.257 | | | <0.001 |
| Yes | 910 (13.98) | 204 (18.39) | 100 (23.24) | | 719 (14.59) | 495 (15.88) | | 833 (12.57) | 381 (23.09) | |
| No | 4999 (86.02) | 804 (81.61) | 289 (76.76) | | 3885 (85.41) | 2207 (84.12) | | 4987 (87.43) | 1105 (76.91) | |
| **Hypertension, n (%)** | | | | <0.001 | | | <0.001 | | | <0.001 |
| Yes | 1598 (26.91) | 353 (33.92) | 151 (39.85) | | 1233 (26.86) | 869 (31.60) | | 1467 (24.33) | 635 (42.13) | |
| No | 4311 (73.09) | 655 (66.08) | 238 (60.15) | | 3371 (73.14) | 1833 (68.40) | | 4353 (75.67) | 851 (57.87) | |
| **Weak/failing kidneys, n (%)** | | | | 0.005 | | | 0.919 | | | 0.001 |
| Yes | 73 (1.27) | 23 (2.06) | 19 (3.63) | | 60 (1.48) | 55 (1.52) | | 66 (1.10) | 49 (2.79) | |
| No | 5836 (98.73) | 985 (97.94) | 370 (96.37) | | 4544 (98.52) | 2647 (98.48) | | 5754 (98.90) | 1437 (97.21) | |

*(Continued)*

**Table 1.** (Continued)

| Variable | Non-depression (n = 5909) | StD (n = 1008) | MDD (n = 389) | P | Non-short sleep (n = 4604) | Short sleep (n = 2702) | P | Non-trouble sleeping (n = 5820) | Trouble sleeping (n = 1486) | P |
|---|---|---|---|---|---|---|---|---|---|---|
| **Cancer/malignancy, n (%)** | | | | 0.276 | | | 0.942 | | | <0.001 |
| Yes | 145 (3.44) | 27 (3.62) | 15 (5.91) | | 110 (3.60) | 77 (3.55) | | 114 (2.76) | 73 (6.28) | |
| No | 5764 (96.56) | 981 (96.38) | 374 (94.09) | | 4494 (96.40) | 2625 (96.45) | | 5706 (97.24) | 1413 (93.72) | |
| **SPS, n (%)** | | | | 0.011 | | | <0.001 | | | 0.337 |
| Negative attitude | 1524 (22.36) | 290 (25.48) | 135 (32.05) | | 1223 (23.59) | 726 (22.63) | | 1526 (22.81) | 423 (24.74) | |
| Neutral attitude | 2572 (46.06) | 457 (48.33) | 143 (42.28) | | 2072 (47.87) | 1100 (42.92) | | 2515 (45.95) | 657 (46.90) | |
| Positive attitude | 1813 (31.58) | 261 (26.19) | 111 (25.67) | | 1309 (28.53) | 876 (34.44) | | 1779 (31.24) | 406 (28.36) | |
| **SED, n (%)** | | | | 0.111 | | | 0.004 | | | 0.004 |
| Q1 | 1116 (17.06) | 225 (22.45) | 92 (19.24) | | 904 (17.63) | 529 (18.39) | | 1136 (17.80) | 297 (18.19) | |
| Q2 | 1348 (23.01) | 192 (19.84) | 87 (23.27) | | 1072 (24.17) | 555 (19.59) | | 1271 (21.94) | 356 (24.75) | |
| Q3 | 1783 (32.92) | 306 (30.16) | 112 (32.88) | | 1366 (32.62) | 835 (32.41) | | 1719 (31.86) | 482 (34.80) | |
| Q4 | 1662 (27.01) | 285 (27.55) | 98 (24.60) | | 1262 (25.57) | 783 (29.62) | | 1694 (28.40) | 351 (22.26) | |

StD, subthreshold depression; MDD, major depressive disorder; SE, standard error; BMI, body mass index; CVD, cardiovascular disease; SPS, sunlight preference score; SED, sunlight exposure duration; M, median; IQR, interquartile range.

of short sleep (aOR = 0.80, 95% CI: 0.65–0.99); While the Q4 demonstrated 22% reduced odds of trouble sleeping (aOR = 0.78, 95% CI: 0.62–0.99), accompanied by a linear trend (P-trend = 0.012). Nonlinearity testing suggested that the association between SED and short sleep was U-shaped (P-nonlinearity = 0.001), and the association between SED and trouble sleeping was inverted U-shaped (P-nonlinearity = 0.018).

The overall fit of the fully adjusted logistic regression models (Model 3) was assessed. The AIC and BIC values for each model are presented in Table 3. Lower values indicate superior model fit. For all outcome variables, the AIC and BIC decreased substantially from the crude (Model 1) to the fully adjusted models (Model 3), demonstrating that the inclusion of demographic, lifestyle, and comorbidity covariates significantly improved model fit (S4 Table). Detailed data are presented in Fig 2 and Tables 2 and 3.

This cross-sectional study identified complex associations between sunlight affinity and male depression/sleep disorders. Higher sunlight affinity correlated with lower depression/trouble sleeping prevalence but higher risk of short sleep in males. The causal relationships and underlying mechanisms need to be elucidated through further investigation, which may be crucial for the development of mental health policies targeting the male population.

## Subgroup analyses by demographic characteristics

Continuous covariates, including age (categorized as 20–39 and 40–59 years) and PIR (<1.3, 1.3–1.5, ≥ 1.5), were transformed into categorical variables for subgroup and interaction analyses. As shown in Fig 3, the inverse associations between SPS and depression/trouble sleeping were more pronounced in widowed/divorced/separated individuals (P-interaction <0.05). Fig 4 highlights the heterogeneity in the associations of SED with depression/short sleep across subgroups. Specifically, significant interaction effects were observed for age, marital status, and PIR (P-interaction <0.05). Among low-income individuals (PIR < 1.3), the inverse association of SED with depression was stronger (P-interaction <0.05). The inverse association between SED and MDD was more pronounced in younger adults (20–39 years), never married individuals, and widowed/divorced/separated subgroups (P-interaction <0.05). Additionally, among those with higher education (>high school) and non-Hispanic Whites, the positive direct association between SED and short sleep was particularly prominent (P-interaction <0.05).

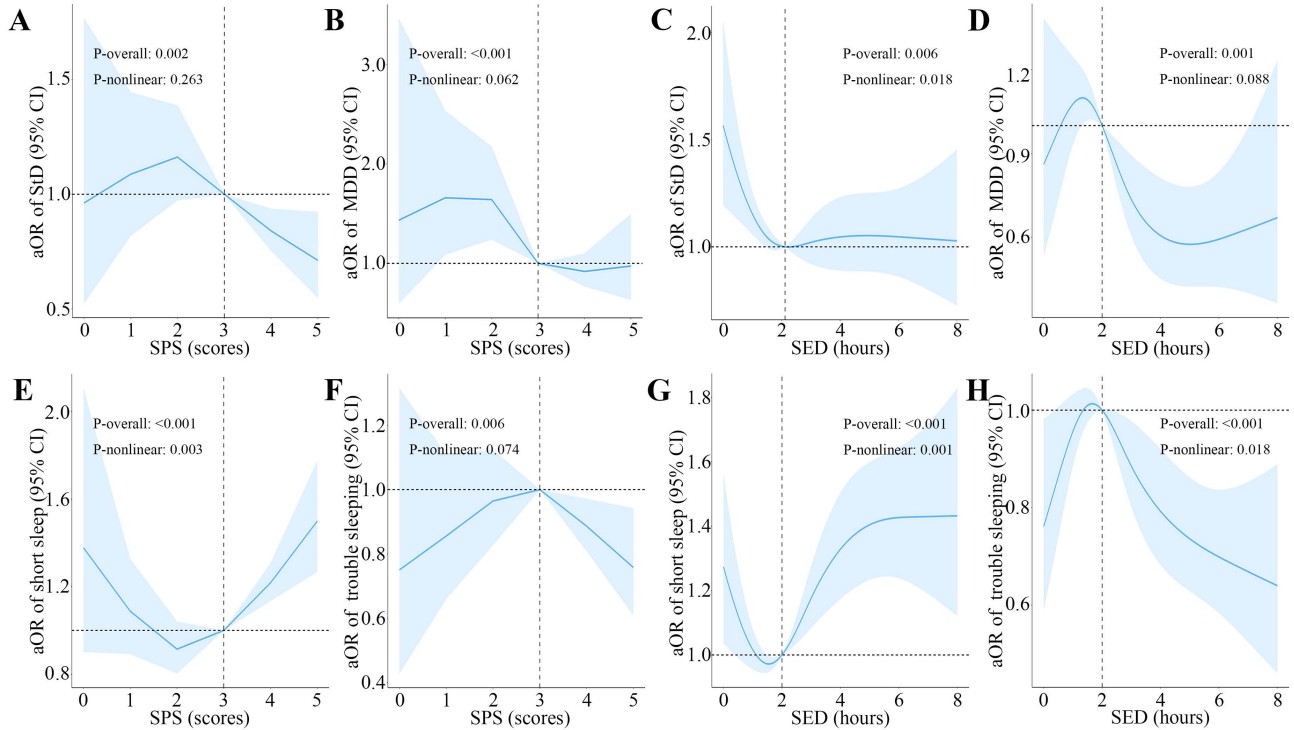

**Fig 2. Restricted cubic spline analyses of the relationships between sunlight affinity and depression and sleep disorders.** SPS, sunlight preference score; SED, sunlight exposure duration; StD, subthreshold depression; MDD, major depressive disorder; aOR, adjusted odds ratio; CI, confidence interval. **(A)** SPS is associated with StD, but no significant nonlinear relationship is observed (P for overall = 0.002; P for non-linear = 0.263). **(B)** SPS is associated with MDD, but no significant nonlinear relationship is observed (P for overall < 0.001; P for non-linear = 0.062). **(C)** SED demonstrates a significant nonlinear association with StD (P for overall = 0.006; P for non-linear = 0.018). **(D)** SED is associated with MDD, but no significant nonlinear relationship is observed (P for overall = 0.001; P for non-linear = 0.088). **(E)** SPS shows a significant nonlinear association with short sleep (P for overall < 0.001; P for non-linear = 0.003). **(F)** SPS is associated with trouble sleeping, but no significant nonlinear relationship is observed (P for overall = 0.006; P for non-linear = 0.074). **(G)** SED shows a significant nonlinear association with short sleep (P for overall < 0.001; P for non-linear = 0.001). **(H)** SED demonstrates a significant nonlinear association with trouble sleeping (P for overall < 0.001; P for non-linear = 0.018). Solid lines represent the aORs, and shaded areas represent the 95% CIs. The reference values were set at the median value of the SPS or SED. All models were adjusted for demographic covariates (including age, race, education, marital status and PIR), lifestyle (including smoking history, alcohol consumption history, physical activity, and BMI) and comorbidities (including CVD, liver condition, asthma, weak/failing kidneys, cancer/malignancy, diabetes, and hypertension).

Subgroup analyses in this cross-sectional study demonstrated heterogeneity in the associations between sunlight affinity and depression/sleep disorders among males across different populations, while maintaining effect direction alignment with the fully adjusted model (Model 3).

### Mediation analyses for sunlight affinity in the depression-sleep disorder relationship

We found that sunlight affinity (IE > 0; P < 0.001; Figs 5A and 5E) might exhibit a suppression effect in the bidirectional negative associations between sleep hours and PHQ-9 scores (DE < 0; TE < 0; P < 0.05), where the IE and DE acted in opposite directions. Specifically, increased sleep hours were associated with lower PHQ-9 scores (TE = $-4.19 \times 10^{-2}$; P = 0.014), and lower sunlight affinity might have exhibited a suppression effect in this association (SPS: 29.98%; SED: 30.85%). Conversely, increased PHQ-9 scores were correlated with reduced sleep hours (TE = $-5.25 \times 10^{-3}$; P = 0.014), a relationship partially suppressed by lower sunlight affinity in a negative direction (SPS: 28.95%; SED: 30.50%). We identified bidirectional positive associations between trouble sleeping and PHQ-9 scores (DE > 0; TE > 0; Figs 5B and 5F). Notably,

**Table 2. Logistic regression model 3 for the association of sunlight affinity with depression and sleep disorders.**

| Variables | | StD | P | MDD | P | Short sleep | P | Trouble sleeping | P |
|---|---|---|---|---|---|---|---|---|---|
| | | aOR (95% CI) | | aOR (95% CI) | | aOR (95% CI) | | aOR (95% CI) | |
| SPS | Scores | 0.88 (0.80–0.96) | 0.007 | 0.80 (0.69–0.92) | 0.003 | 1.11 (1.04–1.19) | 0.002 | 0.92 (0.84–1.01) | 0.074 |
| | Categories | | | | | | | | |
| | Negative attitude | Reference | | Reference | | Reference | | Reference | |
| | Neutral attitude | 0.91 (0.70–1.17) | 0.462 | 0.60 (0.39–0.92) | 0.022 | 1.00 (0.83–1.21) | 0.960 | 0.91 (0.73–1.12) | 0.377 |
| | Positive attitude | 0.69 (0.53–0.92) | 0.011 | 0.52 (0.34–0.79) | 0.003 | 1.34 (1.12–1.62) | 0.002 | 0.85 (0.65–1.11) | 0.235 |
| | P-trend | 0.83 (0.73–0.95) | 0.009 | 0.72 (0.57–0.90) | 0.005 | 1.17 (1.07–1.29) | 0.001 | 0.92 (0.80–1.06) | 0.245 |
| SED | Hours | 0.96 (0.91–1.02) | 0.179 | 0.90 (0.84–0.96) | 0.003 | 1.05 (1.01–1.10) | 0.029 | 0.94 (0.90–0.98) | 0.007 |
| | Categories | | | | | | | | |
| | Q1 | Reference | | Reference | | Reference | | Reference | |
| | Q2 | 0.69 (0.50–0.95) | 0.025 | 0.99 (0.60–1.64) | 0.974 | 0.80 (0.65–0.99) | 0.040 | 1.08 (0.83–1.40) | 0.564 |
| | Q3 | 0.70 (0.51–0.97) | 0.034 | 0.91 (0.61–1.37) | 0.650 | 0.96 (0.80–1.16) | 0.696 | 1.02 (0.83–1.26) | 0.824 |
| | Q4 | 0.70 (0.52–0.94) | 0.022 | 0.68 (0.48–0.96) | 0.033 | 1.08 (0.87–1.35) | 0.470 | 0.78 (0.62–0.99) | 0.044 |
| | P-trend | 0.95 (0.89–1.01) | 0.110 | 0.91 (0.84–0.99) | 0.037 | 1.04 (1.00–1.09) | 0.073 | 0.94 (0.89–0.98) | 0.012 |

SPS, sunlight preference score; SED, sunlight exposure duration; StD, subthreshold depression; MDD, major depressive disorder; aOR, adjusted odds ratio; CI, confidence interval.

Adjusted for demographics, lifestyle, and comorbidities.

**Table 3. Threshold effect analyses for the associations of sunlight affinity with sleep disorders.**

| Outcome | SPS–Short sleep | | SPS–Trouble sleeping | |
|---|---|---|---|---|
| | aOR (95% CI) | P | aOR (95% CI) | P |
| Fitting model by standard linear regression | 1.08 (1.03–1.13) | 0.002 | 0.92 (0.87–0.97) | 0.004 |
| Fitting model by two-piecewise linear regression | | | | |
| Inflection point | 2.867 | | 4 | |
| <Inflection point | 0.98 (0.80–1.19) | 0.823 | 0.95 (0.85–1.05) | 0.332 |
| ≥Inflection point | 1.17 (1.08–1.26) | <0.001 | 0.62 (0.48–0.80) | <0.001 |
| P for likelihood test | | 0.026 | | 0.012 |

SPS, sunlight preference score; SED, sunlight exposure duration; StD, subthreshold depression; MDD, major depressive disorder; aOR, adjusted odds ratio; CI, confidence interval.

Adjusted for demographics, lifestyle, and comorbidities.

in the positive association between trouble sleeping and elevated PHQ-9 scores (TE = 2.50; P < 0.001), diminished sunlight affinity may demonstrate marginal mediation effects, with SPS accounting for 0.28% and SED contributing 0.77% of the mediated proportion.

Further analyses revealed bidirectional positive associations between short sleep and StD (DE > 0; TE > 0; Fig 5C) as well as between trouble sleeping and MDD (DE > 0; TE > 0; Fig 5D). Specifically, short sleep was significantly associated with elevated StD risk (TE = $3.71 \times 10^{-2}$; P < 0.001), with higher SPS could suppress this effect by 4.25%. While StD predicted increased short sleep risk (TE = $6.72 \times 10^{-2}$; P < 0.001), may be partially suppressed by diminished SPS (3.89%). Moreover, the lower SPS demonstrated potential minimal mediation (1.10%) in the positive association between trouble sleeping and higher MDD risk (TE = $1.79 \times 10^{-1}$; P < 0.001). In addition, SED showed a potential suppression effect in the relationship between short sleep and trouble sleeping. Specifically,

| Subgroup Demographic variables | SPS–StD/MDD aOR (95% CI) | | P-interaction | SPS–Short sleep/Trouble sleeping aOR (95% CI) | | P-interaction |
|---|---|---|---|---|---|---|
| **Over all** | 0.88 (0.81 ~ 0.97) | | | 1.11 (1.04 ~ 1.19) | | |
| **Age (year)** | | | 0.338 | | | 0.994 |
| 20–39 | 0.86 (0.77 ~ 0.97) | | | 1.12 (1.03 ~ 1.22) | | |
| 40–59 | 0.90 (0.80 ~ 1.02) | | | 1.11 (1.01 ~ 1.21) | | |
| **Race** | | | 0.580 | | | 0.081 |
| Mexican American | 0.86 (0.73 ~ 1.00) | | | 1.05 (0.94 ~ 1.18) | | |
| Non-Hispanic Black | 0.91 (0.78 ~ 1.06) | | | 1.03 (0.92 ~ 1.14) | | |
| Non-Hispanic White | 0.89 (0.77 ~ 1.04) | | | 1.15 (1.03 ~ 1.29) | | |
| Other Hispanic | 0.77 (0.63 ~ 0.94) | | | 1.00 (0.85 ~ 1.17) | | |
| Other race | 0.96 (0.78 ~ 1.20) | | | 1.19 (1.02 ~ 1.40) | | |
| **Education** | | | 0.643 | | | 0.658 |
| <High school | 0.94 (0.83 ~ 1.06) | | | 1.04 (0.93 ~ 1.17) | | |
| High school | 0.86 (0.74 ~ 1.00) | | | 1.11 (0.95 ~ 1.28) | | |
| >High school | 0.88 (0.77 ~ 1.01) | | | 1.14 (1.04 ~ 1.26) | | |
| **Marital status** | | | 0.033 | | | 0.498 |
| Never married | 0.92 (0.77 ~ 1.08) | | | 1.22 (1.08 ~ 1.38) | | |
| Widowed/Divorced/Separated | 0.72 (0.58 ~ 0.89) | | | 1.17 (0.98 ~ 1.40) | | |
| Married/Living with partner | 0.94 (0.85 ~ 1.04) | | | 1.08 (0.99 ~ 1.16) | | |
| **PIR** | | | 0.687 | | | 0.562 |
| <1.3 | 0.84 (0.75 ~ 0.95) | | | 1.07 (0.97 ~ 1.18) | | |
| 1.3–3.5 | 0.91 (0.80 ~ 1.02) | | | 1.07 (0.97 ~ 1.18) | | |
| ≥3.5 | 0.91 (0.75 ~ 1.11) | | | 1.18 (1.06 ~ 1.32) | | |
| **Over all** | 0.81 (0.70 ~ 0.93) | | | 0.92 (0.84 ~ 1.01) | | |
| **Age (year)** | | | 0.640 | | | 0.506 |
| 20–39 | 0.85 (0.70 ~ 1.03) | | | 0.88 (0.77 ~ 1.01) | | |
| 40–59 | 0.77 (0.65 ~ 0.91) | | | 0.95 (0.85 ~ 1.06) | | |
| **Race** | | | 0.183 | | | 0.798 |
| Mexican American | 1.06 (0.83 ~ 1.35) | | | 0.96 (0.82 ~ 1.12) | | |
| Non-Hispanic Black | 0.95 (0.73 ~ 1.23) | | | 0.90 (0.78 ~ 1.03) | | |
| Non-Hispanic White | 0.73 (0.60 ~ 0.89) | | | 0.90 (0.80 ~ 1.02) | | |
| Other Hispanic | 0.85 (0.62 ~ 1.15) | | | 1.00 (0.80 ~ 1.24) | | |
| Other race | 0.78 (0.55 ~ 1.12) | | | 0.98 (0.82 ~ 1.17) | | |
| **Education** | | | 0.303 | | | 0.089 |
| <High school | 0.90 (0.74 ~ 1.08) | | | 0.87 (0.77 ~ 0.99) | | |
| High school | 0.69 (0.52 ~ 0.91) | | | 0.81 (0.70 ~ 0.94) | | |
| >High school | 0.85 (0.68 ~ 1.06) | | | 0.98 (0.87 ~ 1.10) | | |
| **Marital status** | | | 0.021 | | | 0.034 |
| Never married | 0.75 (0.54 ~ 1.05) | | | 0.90 (0.74 ~ 1.09) | | |
| Widowed/Divorced/Separated | 0.62 (0.44 ~ 0.88) | | | 0.75 (0.63 ~ 0.89) | | |
| Married/Living with partner | 0.98 (0.82 ~ 1.17) | | | 0.99 (0.88 ~ 1.12) | | |
| **PIR** | | | 0.624 | | | 0.257 |
| <1.3 | 0.79 (0.65 ~ 0.97) | | | 0.95 (0.83 ~ 1.08) | | |
| 1.3–3.5 | 0.79 (0.66 ~ 0.94) | | | 0.85 (0.76 ~ 0.94) | | |
| ≥3.5 | 0.90 (0.61 ~ 1.33) | | | 0.96 (0.82 ~ 1.11) | | |

**Fig 3. Subgroup analyses of the associations between sunlight preference score and depression and sleep disorders.** SPS, sunlight preference score; StD, subthreshold depression; MDD, major depressive disorder; aOR, adjusted odds ratio; CI, confidence interval; PIR, poverty income ratio. Adjusted for demographics, lifestyle, and comorbidities, except for the stratification factor itself.

short sleep was associated with elevated trouble sleeping risk (TE = $3.41 \times 10^{-2}$; P < 0.001), with higher SED linked to a reduction in this association by 13.12%. Conversely, trouble sleeping was associated with increased short sleep risk (TE = $4.27 \times 10^{-2}$; P < 0.001), and lower SED was linked to a reduction in this association by 12.87%. See S5 Table for more information.

Mediation analysis revealed the complex significance of sunlight affinity in the cyclical relationship between male depression and sleep disorders, which provided valuable insights for further exploring the target research of sunlight affinity on the comorbidity of depression and sleep disorders among males.

| Subgroup | SED–StD/MDD | | SED–Short sleep/Trouble sleeping | |
|---|---|---|---|---|
| Demographic variables | aOR (95% CI) | P-interaction | aOR (95% CI) | P-interaction |
| **Over all** | 0.96 (0.91 ~ 1.02) | | 1.05 (1.00 ~ 1.10) | |
| **Age (year)** | | 0.068 | | 0.734 |
| 20–39 | 0.94 (0.88 ~ 1.02) | | 1.05 (0.99 ~ 1.11) | |
| 40–59 | 1.00 (0.92 ~ 1.07) | | 1.05 (0.99 ~ 1.11) | |
| **Race** | | 0.454 | | 0.003 |
| Mexican American | 0.91 (0.84 ~ 0.97) | | 0.97 (0.91 ~ 1.03) | |
| Non-Hispanic Black | 1.04 (0.96 ~ 1.14) | | 1.02 (0.95 ~ 1.08) | |
| Non-Hispanic White | 0.96 (0.89 ~ 1.04) | | 1.09 (1.02 ~ 1.16) | |
| Other Hispanic | 0.97 (0.85 ~ 1.12) | | 0.94 (0.85 ~ 1.03) | |
| Other race | 0.97 (0.82 ~ 1.15) | | 1.05 (0.94 ~ 1.18) | |
| **Education** | | 0.359 | | 0.020 |
| <High school | 1.01 (0.91 ~ 1.11) | | 0.95 (0.89 ~ 1.00) | |
| High school | 0.92 (0.84 ~ 1.01) | | 1.03 (0.97 ~ 1.10) | |
| >High school | 0.97 (0.90 ~ 1.04) | | 1.10 (1.03 ~ 1.17) | |
| **Marital status** | | 0.707 | | 0.594 |
| Never married | 0.98 (0.88 ~ 1.09) | | 1.04 (0.97 ~ 1.11) | |
| Widowed/Divorced/Separated | 0.96 (0.86 ~ 1.07) | | 1.09 (0.98 ~ 1.21) | |
| Married/Living with partner | 0.96 (0.90 ~ 1.02) | | 1.04 (0.99 ~ 1.09) | |
| **PIR** | | 0.013 | | 0.412 |
| <1.3 | 0.89 (0.83 ~ 0.96) | | 1.05 (1.00 ~ 1.11) | |
| 1.3–3.5 | 1.04 (0.97 ~ 1.12) | | 1.02 (0.97 ~ 1.08) | |
| ≥3.5 | 0.91 (0.81 ~ 1.03) | | 1.08 (1.00 ~ 1.17) | |
| **Over all** | 0.91 (0.85 ~ 0.97) | | 0.94 (0.91 ~ 0.99) | |
| **Age (year)** | | 0.026 | | 0.718 |
| 20–39 | 0.85 (0.79 ~ 0.93) | | 0.94 (0.88 ~ 1.00) | |
| 40–59 | 0.97 (0.88 ~ 1.07) | | 0.95 (0.90 ~ 1.01) | |
| **Race** | | 0.735 | | 0.581 |
| Mexican American | 0.96 (0.82 ~ 1.12) | | 0.90 (0.84 ~ 0.98) | |
| Non-Hispanic Black | 0.81 (0.71 ~ 0.92) | | 0.91 (0.84 ~ 0.99) | |
| Non-Hispanic White | 0.91 (0.83 ~ 1.01) | | 0.96 (0.90 ~ 1.01) | |
| Other Hispanic | 0.86 (0.72 ~ 1.01) | | 0.92 (0.82 ~ 1.03) | |
| Other race | 0.84 (0.69 ~ 1.02) | | 0.92 (0.81 ~ 1.05) | |
| **Education** | | 0.342 | | 0.207 |
| <High school | 0.91 (0.80 ~ 1.03) | | 1.02 (0.92 ~ 1.13) | |
| High school | 0.81 (0.71 ~ 0.93) | | 0.89 (0.83 ~ 0.96) | |
| >High school | 0.96 (0.85 ~ 1.08) | | 0.96 (0.90 ~ 1.02) | |
| **Marital status** | | 0.024 | | 0.314 |
| Never married | 0.79 (0.65 ~ 0.95) | | 0.98 (0.90 ~ 1.07) | |
| Widowed/Divorced/Separated | 0.84 (0.74 ~ 0.95) | | 0.93 (0.85 ~ 1.01) | |
| Married/Living with partner | 0.99 (0.90 ~ 1.09) | | 0.94 (0.88 ~ 1.00) | |
| **PIR** | | 0.035 | | 0.245 |
| <1.3 | 0.79 (0.70 ~ 0.89) | | 1.01 (0.93 ~ 1.08) | |
| 1.3–3.5 | 1.00 (0.90 ~ 1.11) | | 0.94 (0.88 ~ 1.00) | |
| ≥3.5 | 0.96 (0.78 ~ 1.19) | | 0.92 (0.84 ~ 1.00) | |

**Fig 4. Subgroup analyses of the associations between sunlight exposure duration and depression and sleep disorders.** SED, sunlight exposure duration; StD, subthreshold depression; MDD, major depressive disorder; aOR, adjusted odds ratio; CI, confidence interval; PIR, poverty income ratio. Adjusted for demographics, lifestyle, and comorbidities, except for the stratification factor itself.

## Sensitivity analyses for sunlight affinity with depression and sleep disorders

First, we excluded participants who might have been unwilling to be exposed to sunlight due to potential health issues (n = 46, 0.63%) or who might have been unable to go outdoors due to time constraints (n = 418, 5.72%), resulting in a total exclusion of 431 participants (5.90%). Subsequently, data from the remaining 6,875 participants were re-analyzed using multivariate logistic regression, RCS analysis, and threshold effect analysis with the same covariate adjustment strategy as in the primary analysis to evaluate the sensitivity of the results.

Sensitivity analyses showed that each score gain in SPS demonstrated a 15% reduction of StD risk (aOR = 0.85, 95% CI: 0.76–0.94) as well as a 22% reduction of MDD risk (aOR = 0.78, 95% CI: 0.66–0.91) but a 13% increase in the

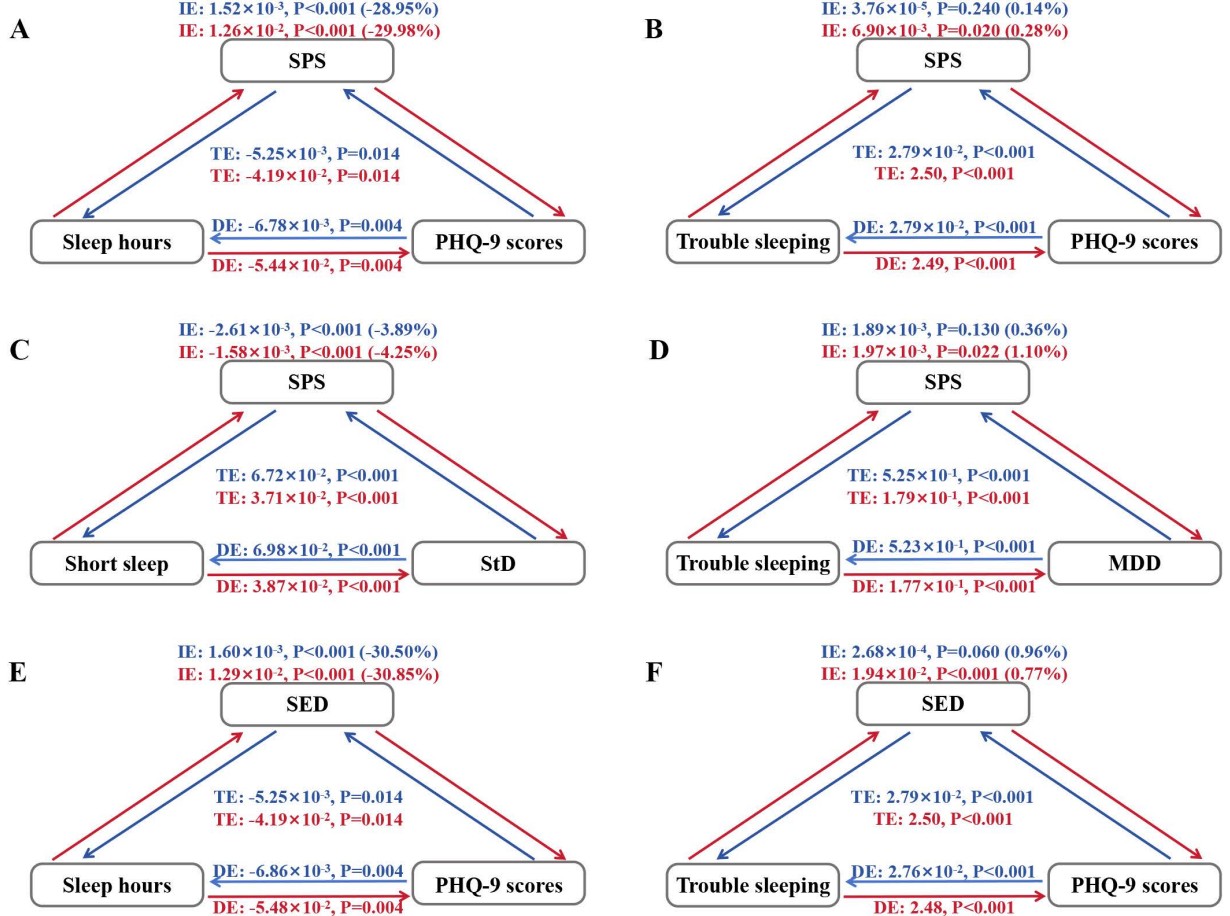

**Fig 5. Mediation analyses of sunlight affinity on the associations between depression and sleep disorders.** SPS, sunlight preference score; SED, sunlight exposure duration; StD, subthreshold depression; MDD, major depressive disorder. Indirect effect (IE) + Direct effect (DE) = Total effect (TE), IE/TE × 100% = Proportion of mediation (in parentheses). Red arrows and text indicate left-to-right mediating associations, while blue arrows and text indicate right-to-left mediating associations. **(A)** Mediation effect of SPS on sleep hours and PHQ-9 scores. **(B)** Mediation effect of SPS on trouble sleeping and PHQ-9 scores. **(C)** Mediation effect of SPS on short sleep and StD. **(D)** Mediation effect of SPS on trouble sleeping and MDD. **(E)** Mediation effect of SED on sleep hours and PHQ-9 scores. **(F)** Mediation effect of SED on trouble sleeping and PHQ-9 scores. Adjusted for demographics, lifestyle, and comorbidities.

risk of short sleep (aOR = 1.13 95% CI: 1.06–1.20). A potential nonlinear association between SPS and short sleep was suggested by RCS analysis (P-nonlinear = 0.016). The threshold effect analysis further identified a critical inflection point at SPS ≥ 3, where the risk of short sleep elevated considerably (aOR = 1.17, 95% CI: 1.08–1.27), whereas SPS ≥ 4 was linked to reduced trouble sleeping risk (aOR = 0.62, 95% CI: 0.48–0.81). Moreover, a 10% decrease in MDD risk was observed for each hour increase in SED (aOR = 0.90, 95% CI: 0.83–0.98), along with a 7% decrease in trouble sleeping risk (aOR = 0.93, 95% CI: 0.89–0.98), while elevated short sleep risk by 6% (aOR = 1.06, 95% CI: 1.02–1.11). These findings demonstrated concordance with the primary analysis, except that the original nonlinear associations between SED and StD/sleep disorders were attenuated (P-nonlinear >0.05). Complete results are detailed in S6 and S7 Tables, S1 Fig.

## Discussion

This cross-sectional investigation of 7,306 adult males from NHANES (2009–2020) revealed multifaceted associations between sunlight affinity and male mental health outcomes. Sunlight affinity was inversely associated with depression and

trouble sleeping and positively associated with short sleep in males. These associations persisted consistently across demographic strata, with more pronounced inverse relationships with MDD observed in males experiencing marital dissolution. Furthermore, the potential suppression effect of sunlight affinity in the male depression–sleep disorders cycle suggests that sunlight affinity may hold significant reference value for research on mental health policies targeting male populations.

This study revealed potential inverse association between sunlight affinity and depression. The inverse association between SPS and depression might be explained by dual biological and psychophysiological mechanisms. From a biological perspective, sunlight-seeking behavior aligns with innate human tendencies. A multinational survey involving 20 countries found that 83.2% of participants expressed voluntary sun-seeking intentions [14]. Individuals inherently prefer environments with windows that provide access to daylight in their daily lives [29]. Genomic analyses of over 260,000 individuals demonstrated that alleles such as CADM2 and TMEM182 regulate phototactic behaviors via vitamin D pathways, thereby biologically underpinning sunlight-seeking instincts [30]. Psychologically, sunlight fosters positive emotional states that counteract depressive pathology. Sunlit environments enhance well-being and stress resilience by promoting relaxation and alleviating anxiety [31]. These effects are mediated by dopamine and endorphin-driven neurochemical pathways, which stabilize hypothalamic-pituitary-adrenal axis function and mitigate anxiety-depression cascades [32]. Conversely, light-averse individuals display emotional dysregulation, with heightened impulsivity during extreme emotional states [33]. Additionally, the positive correlation between SPS and SED suggests that a positive sunlight preference is frequently linked to longer SED, and numerous studies have demonstrated that individuals who receive more sunlight exposure have a lower risk of depression [33,34]. Mechanistically, sunlight exposure correlate with lower depression risk through six potential pathways: Vitamin D synthesis [35], upregulation of anti-inflammatory cytokines [36], melatonin suppression [37], serotonin modulation via 5-HT1A receptors [38], cortisol inhibition [39], and amygdala regulation via ipRGCs [40].

We identified a significant association between sunlight affinity and short sleep risk aligning with the findings of Elovainio et al. which linked prolonged sunlight exposure to reduced sleep hours [41]. Beyond sunlight exposure behavior, which constitutes one of the two dimensions of sunlight affinity, the subjective propensity for sunlight exposure (SPS) was found to be associated with a concurrent elevation in short sleep and a decline in trouble sleeping when SPS reached a specific threshold (≥4). This suggests that the reduction in sleep duration may not be wholly negative but could be offset by improvements in sleep quality and consolidation, potentially mediated by stronger circadian entrainment. It is plausible that individuals with high sunlight affinity, often engaged in active outdoor lifestyles, may trade some sleep opportunity for these activities yet benefit from the sleep-promoting effects of daytime light exposure. Therefore, this phenomenon suggests that high sunlight affinity may be associated with better sleep quality, but prospective clinical studies are needed to validate this possibility. Mechanistically, ipRGCs relay light signals to the suprachiasmatic nucleus through the retinohypothalamic tract, mediating circadian synchronization [42]. This pathway optimizes sleep-wake transitions and enhances daytime alertness by coordinating melatonin suppression and cortisol rhythmicity. Moreover, ipRGCs modulate emotional processing through connections to limbic structures, such as the central amygdala, establishing bidirectional interactions among light exposure, affective states, and sleep quality [40,42].

Subgroup analyses revealed heterogeneity in sunlight affinity across populations. Specifically, SED exhibited stronger inverse correlations with depression in low-income individuals, which may be linked to heightened depression susceptibility associated with financial stress [43]. The low-cost and accessible nature of sunlight exposure highlights its potential as a resource for economically disadvantaged groups. Never married status and marital disruptions (widowhood, divorce, or separation) were linked to an elevated likelihood of depression and sleep disorders, potentially attributable to reduced social support [44,45]. However, elevated sunlight affinity could help compensate for this deficit, potentially through fostering proactive lifestyle attitudes [46]. Males aged 20–40 years are prone to elevated psychosocial stressors, including occupational competition and familial responsibilities. Moderate outdoor sunlight exposure may alleviate these stressors

by enhancing psychological and physiological relaxation [47]. Furthermore, race and educational level modified the relationship between SED and short sleep. Non-Hispanic White individuals may be more likely to engage in outdoor activities or reside in sunnier regions [48], potentially increasing sunlight exposure duration. Their less-pigmented skin heightens sensitivity to sunlight, potentially amplifying its physiological effects [49]. Higher education levels are linked to greater health awareness and behaviors, including the recognition of the significance of sun exposure with respect to vitamin D synthesis as well as overall health. This may lead to more active sun-seeking practices [50].

Mediation analyses revealed bidirectional associations between depression and sleep disorders in males, consistent with the neurobehavioral pathways proposed by Raza et al [51]. Sunlight affinity may have complex implications in these relationships, though some have modest effect sizes. Notably, threshold effects possibly exist in the complex interplay among psychological, sleep, and psychiatric disorders, warranting further investigation. Nevertheless, integrating our prior multivariate regression results, we identified several key insights. Firstly, in the cycle of short sleep and StD, the positive sunlight preference was associated with reduced StD risk in short sleep individuals, potentially by compensating for circadian disruption caused by insufficient sleep [52]. Conversely, moderate sunlight preference might be linked to lower short sleep risk in StD patients, possibly through mood stabilization mechanisms [53]. Secondly, in the cycle of trouble sleeping and MDD, while poor sleep quality is a well-established risk factor for MDD [54], our data suggest that sunlight affinity may be relevant to this bidirectional relationship, possibly through enhancing serotonin synthesis or circadian alignment in vulnerable males. Thirdly, in the cycle of short sleep and trouble sleeping, the suppression effect of by SED suggests that increased daylight exposure could attenuate bidirectional short sleep–trouble sleeping relationships (12.87–13.12% suppression), likely via circadian regulation [55]. These findings suggest that future interventions exploring sunlight affinity, particularly in high-risk subgroups, might offer a potential dual-benefit strategy to simultaneously address sleep disorders and depression, potentially disrupting maladaptive feedback loops between these conditions. For instance, light exposure interventions tailored to circadian timing (e.g., morning bright light therapy) may enhance both sleep quality and mood regulation, as evidenced by prior trials linking light exposure to improved memory consolidation and emotional resilience [56]. However, these results should be interpreted cautiously and require further validation through prospective studies to substantiate causality and elucidate potential pathways.

In sensitivity analyses, the overall significant associations persisted, thereby reinforcing the robustness of the association of sunlight affinity with depression and sleep disorders. From a localized perspective, the original nonlinear associations between SED and StD/sleep disorders were attenuated, which may reflect the unique characteristics of the excluded population. For instance, a prior study found that operating room nurses working extended hours in sunless environments exhibit poorer mental and sleep health compared to the general population [57]. Therefore, the excluded population possibly demonstrates a stronger association with StD/sleep disorders relative to other groups. Their exclusion homogenized the sample, reducing local variability and aligning the association closer to a linear trend [58]. However, as an exploratory study aiming to characterize associations across the general population with depression/sleep disorders, we retained all participants without excluding extreme values, as such exclusions would narrow the applicability of study results to populations with unavoidable sunlight avoidance. The underlying mechanisms should be further clarified through stratified analyses conducted on extreme value populations in future studies.

This study has several strengths. Notably, we innovatively developed a dual-dimensional sunlight affinity assessment indicator that integrates psychological (SPS) and behavioral (SED) metrics, providing a complementary perspective on participants' affinity for sunlight from different angles, thereby transcending traditional single-factor paradigms. Our cross-sectional study, which used population-weighted estimates from the nationally representative NHANES data (cycles spanning 2009–2020), focused on depression and sleep disorders in males, tackling the scarcity of research in this domain. The threshold effects between sunlight affinity and sleep disorders highlight modifiable lifestyle factors that could inform future intervention research in male sleep disorders. Given the suboptimal treatment situation among males with MDD [59], the association between sunshine affinity and StD may provide new insights for mental health

prevention strategies for male MDD. Nevertheless, acknowledgement should be made of some limitations. First, As the cross-sectional design of this study inherently precludes the establishment of causal relationships, there is a need for longitudinal designs, particularly prospective cohort studies, to confirm these associations. Second, despite adjusting for multiple covariates, residual confounding from unmeasured environmental factors (e.g., detailed geographic location, seasonal and daily variations in sunlight intensity and duration, air pollution) may have influenced the results. The lack of this data is a limitation, and their inclusion in future research could help clarify the observed associations. Third, NHANES does not release detailed occupational codes that would allow us to directly classify participants into outdoor versus indoor jobs. Consequently, we could not explore whether the observed associations differ between men with occupational sunlight exposure and those primarily working indoors. Future cycles of NHANES or linked occupational databases could help address this question. Fourth, our measure of sunlight exposure (SED) captured only exposure to natural sunlight during daytime hours. We did not have data on exposure to artificial light sources, such as bright light therapy lamps or full-spectrum lighting, which are commonly used, particularly in northern latitudes or for treating seasonal affective disorders. The effects of such artificial light sources on depression and sleep disorders may differ from those of natural sunlight and could represent a potential confounding factor not accounted for in our analysis. Fifth, our models did not adjust for specific clinical sleep disorders such as obstructive sleep apnea, restless legs syndrome, or narcolepsy, which are strong determinants of sleep duration and quality. The NHANES database is deficient in systematic data regarding these clinical diagnoses for all participants. Although we adjusted for key risk factors and comorbidities like BMI, HTN, and CVD, which are proxies for conditions like obstructive sleep apnea, residual confounding remains a possibility. Sixth, our findings are not generalizable to males aged 60 years and older, as the NHANES protocol for the cycles used did not collect data on sunlight affinity (SPS and SED) in this age group. This is a significant limitation, given the high prevalence of depression and sleep disorders among older adults. The associations between sunlight affinity and mental health outcomes may differ in older populations due to factors such as retirement, altered circadian rhythms, increased chronic disease burden, and reduced mobility. Future studies specifically designed to investigate these relationships in older adult populations are warranted. Finally, variables based on self-reports (e.g., SED, sleep hours) may be influenced by recall bias, future research should utilize objective methods (e.g., actigraphy, polysomnography) to validate these findings.

## Conclusion

This study revealed that sunlight affinity was inversely associated with depression and trouble sleeping and positively associated with short sleep in males. Further longitudinal studies are needed to confirm causality.

## Supporting information

**S1 Table. Pearson correlation matrix between sunlight exposure duration and sunlight preference score.**
N = sample size.
(DOCX)

**S2 Table. Univariate linear regression analysis of sunlight preference score and sunlight exposure duration.**
CI: Confidence Interval.
(DOCX)

**S3 Table. Logistic regression models for the association of sunlight affinity with depression and sleep disorders.**
SPS, sunlight preference score; SED, sunlight exposure duration; StD, subthreshold depression; MDD, major depressive disorder; OR, odds ratio; aOR, adjusted odds ratio; CI, confidence interval. Model 1 was unadjusted. Model 2 was adjusted for demographics.
(DOCX)

**S4 Table. Goodness-of-fit statistics for the logistic regression models.** SPS, sunlight preference score; SED, sunlight exposure duration; StD, subthreshold depression; MDD, major depressive disorder; AIC, Akaike information criterion; BIC, Bayesian information criterion. Model 1 was without adjustment for covariates. Model 2 adjusted for demographic covariates (including age, race, education, marital status and PIR). Model 3 further adjusted for lifestyle (including smoking history, alcohol consumption history, physical activity, and BMI) and comorbidities (including CVD, liver condition, asthma, weak/failing kidneys, cancer/malignancy, diabetes, and hypertension) based on Model 2.
(DOCX)

**S5 Table. Mediation analyses results.** SPS, sunlight preference score; SED, sunlight exposure duration; StD, subthreshold depression; MDD, major depressive disorder; CI, confidence interval. Adjusted for demographics, lifestyle, and comorbidities.
(DOCX)

**S6 Table. Logistic regression results after exclusion of extreme values.** SPS, sunlight preference score; SED, sunlight exposure duration; StD, subthreshold depression; MDD, major depressive disorder; OR, odds ratio; aOR, adjusted odds ratio; CI, confidence interval. Adjusted for demographics, lifestyle, and comorbidities.
(DOCX)

**S7 Table. Threshold effect analysis results after exclusion of extreme values.** SPS, sunlight preference score; SED, sunlight exposure duration; StD, subthreshold depression; MDD, major depressive disorder; OR, odds ratio; aOR, adjusted odds ratio; CI, confidence interval. Adjusted for demographics, lifestyle, and comorbidities.
(DOCX)

**S1 Fig. Restricted cubic spline analysis results after exclusion of extreme values.** SPS, sunlight preference score; SED, sunlight exposure duration; StD, subthreshold depression; MDD, major depressive disorder; aOR, adjusted odds ratio; CI, confidence interval. (A) SPS is associated with StD, but no significant nonlinear relationship is observed. (B) SPS is associated with MDD, but no significant nonlinear relationship is observed. (C) SED is associated with StD, but no significant nonlinear relationship is observed. (D) SED is associated with MDD, but no significant nonlinear relationship is observed. (E) SPS shows a significant nonlinear association with short sleep. (F) SPS is associated with trouble sleeping, but no significant nonlinear relationship is observed. (G) SED is associated with short sleep, but no significant nonlinear relationship is observed. (H) SED is associated with trouble sleeping, but no significant nonlinear relationship is observed. Adjusted for demographics, lifestyle, and comorbidities.
(TIF)

## Acknowledgments

All authors extend their sincere gratitude to the crew and participants of the NHANES study for their valuable efforts and excellent cooperation.

## Author contributions

**Conceptualization:** Haifeng Liu, Jia Yang, Tiejun Liu.

**Data curation:** Haifeng Liu, Jia Yang.

**Formal analysis:** Tiejun Liu, Weimin Zhao.

**Methodology:** Haifeng Liu, Jia Yang, Weimin Zhao.

**Supervision:** Jia Yang.

**Writing – original draft:** Haifeng Liu.

**Writing – review & editing:** Jia Yang.

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
