## [Decision Letter · Decision Letter 0]

15 Sep 2025

Dear Dr. Yang,

Thank you for submitting your manuscript to PLOS ONE. After careful consideration, we feel that it has merit but does not fully meet PLOS ONE’s publication criteria as it currently stands. Therefore, we invite you to submit a revised version of the manuscript that addresses the points raised during the review process.

We look forward to receiving your revised manuscript.

Kind regards,

Mohammad Hossein Ebrahimi

Academic Editor

PLOS ONE

Reviewers' comments:

Reviewer's Responses to Questions

**Comments to the Author**

1. Is the manuscript technically sound, and do the data support the conclusions?

Reviewer #1: Yes

Reviewer #2: Yes

Reviewer #3: Yes

2. Has the statistical analysis been performed appropriately and rigorously?

Reviewer #1: Yes

Reviewer #2: Yes

Reviewer #3: Yes

3. Have the authors made all data underlying the findings in their manuscript fully available?

Reviewer #1: Yes

Reviewer #2: Yes

Reviewer #3: Yes

4. Is the manuscript presented in an intelligible fashion and written in standard English?

Reviewer #1: Yes

Reviewer #2: Yes

Reviewer #3: Yes

Reviewer #1: Thank you for allowing me to review your manuscript. I feel like although the relationship between sleep and depression is known already, the important addition you have included is the subjective sunlight affinity based on a self-reports and as you rightly mentioned in your discussion, this could mean a lot of biases and difficult to be certain about its accuracy. Was there data about the seasons/ sunlight per day in that location the subjects were selected from? What is your conclusion about the fact that short sleep is related to sun exposure? Was there a different between men who worked outdoor jobs vs indoor jobs?

Reviewer #2: This is a robust cross sectional study evaluating an important topic for public health. I propose a few questions.

1. I suspect this is evaluating natural sunlight and not a sun lamp or other artificial sun. You way add a clear statement on this as many who live in northern parts of the country regularly use artificial sun lamps.

2. OSA is mentioned in the introduction but no part of the modeling for the logistic regression. Other sleep disorders such as OSA, particularly untreated OSA, restless legs, narcolepsy, etc. would have a large impact on the length and quality of sleep.

3. Why was the maximum age 59 years? Older individuals experience high rates of depression in this group and may change the data.

Reviewer #3: The manuscript could be further improved.

Line 97: The reference number of approvals is to be stated.

Line 145-149: It is unclear how the sleep hours were derived. More information is to be provided.

Line 167: Q1, Q3 is to be replaced with IQR

OR to be replaced with aOR, where possible in the text/tables, to differentiate crude and adjusted.

The fulfilment of statistical tests assumptions is to be stated e,g, multicollinearity etc

Model fit for each statistical test, where applicable, is to be presented.

Line 260: 20–40 and 40–60 years categories seem overlap.

Figure 2: Legend to be provided and more detailed information is to be provided, e.g., model details, statistical test for p non-linear, etc.

Line 178: Likelihood tests to be replaced with likelihood ratio tests. Likewise with S6Table.

Line 184 DE; symbol ; is to be replaced with semi-colon :

Line 189: Model fit to be mentioned and presented in the results section.

S3 Table: typo withdepression

S4 Table: The decimal points are to be standardized.

Some references do not conform to the journal’s required format

**Do you want your identity to be public for this peer review?** For information about this choice, including consent withdrawal, please see our Privacy Policy

Reviewer #1: No

Reviewer #2: No

Reviewer #3: No

---

## [Author Response · Author response to Decision Letter 1]

21 Sep 2025

Response Letter

Dear PLOS ONE Editorial Team,

We sincerely appreciate the opportunity to submit this revised manuscript entitled ”Associations of sunlight affinity with depression and sleep disorders in American males: Evidence from NHANES 2009-2020” (PONE-D-25-14823) to PLOS ONE. We extend our deepest gratitude to the editorial team and reviewers for their substantial time investment and insightful critiques, which have significantly strengthened our work.

All reviewer comments have been meticulously addressed through comprehensive revisions. Below, we present the reviewers' comments in italics, followed by our point-by-point responses in blue text (with key modifications highlighted in red). Supporting documentation appears in standard font, while manuscript changes are tracked using yellow highlighting.

The detailed responses follow this section:

Responses to Comments of Reviewer #1:

Comment 1: Was there data about the seasons/ sunlight per day in that location the subjects were selected from?

Response:

We thank the reviewer for raising this crucial point regarding the potential influence of geographical and seasonal variations in sunlight availability. We acknowledge that this is an important limitation of our study. However, the NHANES database does not provide detailed meteorological data (e.g., daily sunlight hours, UV index) for each participant's location or the specific season of their interview. This limitation is inherent to the dataset and is a common challenge in large-scale national surveys.

To address this potential confounding factor as much as possible within the constraints of the available data, we took the following steps:

Our assessment of Sunlight Exposure Duration (SED) was based on participants' self-reported average time spent outdoors over the previous 30 days. This period-based measure helps to mitigate the impact of day-to-day weather fluctuations and captures a more habitual behavior pattern.

The consistency of our main findings in the sensitivity analyses, even after excluding participants with extreme values, suggests that the observed associations are robust and not solely driven by unmeasured seasonal outliers.

We recognize that the lack of granular environmental data may have influenced our results. We have explicitly acknowledged this limitation in the Discussion section of our manuscript and geo-coded meteorological data are warranted to confirm our findings and elucidate the precise role of seasonal and geographical sunlight variations.

Revised Manuscript with Track Changes (lines 439-442):

“Second, despite adjusting for multiple covariates, residual confounding from unmeasured environmental factors (e.g., detailed geographic location, seasonal and daily variations in sunlight intensity and duration, air pollution) may have influenced the results. The lack of this data is a limitation, and their inclusion in future research could help clarify the observed associations.”

Comment 2: What is your conclusion about the fact that short sleep is related to sun exposure?

Response:

We appreciate the reviewer for this insightful question regarding the positive association between sunlight exposure duration (SED) and short sleep, which indeed appears counterintuitive at first glance. Our conclusion, based on the totality of our findings, is that this relationship likely reflects a combination of behavioral patterns and physiological effects, and it may not be entirely negative. We interpret this association within a framework that distinguishes between sleep quantity (duration) and sleep quality (e.g., trouble sleeping). Our key observations are:

Behavioral mechanism: Individuals with longer SED likely engage in more outdoor activities (e.g., work, exercise, socializing), which may compress available time for sleep, leading to a reduction in total sleep hours. This represents a socio-behavioral link where lifestyle choices associated with high sun exposure could incidentally limit sleep opportunity.

Physiological mechanism and a more nuanced view: Crucially, our data suggest that while increased sun exposure is associated with shorter sleep duration, it is simultaneously associated with better sleep quality. This is strongly evidenced by the negative association we found between SED and trouble sleeping (OR = 0.94, 95% CI: 0.90–0.98). This indicates that although individuals with high sunlight affinity may sleep for a shorter period, they are less likely to suffer from sleep disturbances and insomnia symptoms. Furthermore, the non-linear (U-shaped) relationship we identified between SPS (sunlight preference score)/short sleep and SED/short sleep suggests the relationship is complex and not simply linear. The positive association manifests after a certain threshold, which may represent a point where the circadian-phase-shifting effects of light become significant.

Integration of dimensions: The positive correlation between SPS (psychological preference) and SED (behavioral exposure) suggests that individuals who seek out sunlight are also those who spend more time in it. This high sunlight affinity may reinforce a robust circadian rhythm through the ipRGC-SCN pathway [1]. This enhanced circadian entrainment could lead to more consolidated and efficient sleep, potentially allowing for adequate rest within a shorter time frame, thereby reducing the perceived need for longer sleep and decreasing the risk of trouble sleeping.

In conclusion, we do not interpret the positive association between sunlight affinity and short sleep as a purely detrimental effect. Instead, we propose that it may represent a lifestyle phenotype where sufficient daytime light exposure promotes circadian health and sleep quality, potentially compensating for a modest reduction in sleep quantity. This hypothesis that sunlight affinity might be linked to more efficient rather than merely deficient sleep needs to be directly tested in future research employing objective measures of both sleep duration and sleep architecture (e.g., actigraphy, polysomnography).

The explanation regarding "short sleep is related to sun exposure" has been revised in the "Discussion" section.

Revised Manuscript with Track Changes (lines 375-382):

“Beyond sunlight exposure behavior, which constitutes one of the two dimensions of sunlight affinity, the subjective propensity for sunlight exposure (SPS) was found to be associated with a concurrent elevation in short sleep and a decline in trouble sleeping when SPS reached a specific threshold (≥4). This suggests that the reduction in sleep duration may not be wholly negative but could be offset by improvements in sleep quality and consolidation, potentially mediated by stronger circadian entrainment. It is plausible that individuals with high sunlight affinity, often engaged in active outdoor lifestyles, may trade some sleep opportunity for these activities yet benefit from the sleep-promoting effects of daytime light exposure. Therefore, this phenomenon suggests that high sunlight affinity may be associated with better sleep quality, but prospective clinical studies are needed to validate this possibility.”

Reference

1. Mahoney HL, Schmidt TM. The cognitive impact of light: illuminating ipRGC circuit mechanisms. Nat Rev Neurosci. 2024;25: 159–175. doi:10.1038/s41583-023-00788-5

Comment 3: Was there a different between men who worked outdoor jobs vs indoor jobs?

Response:

We extend our gratitude to the reviewer for highlighting this critical issue. The distinction between outdoor and indoor occupations is indeed a relevant factor that could influence an individual's sunlight exposure patterns and, consequently, the study outcomes. However, the public-use NHANES 2009–2020 files do not contain a variable that directly distinguishes “outdoor” from “indoor” occupations. Therefore, we could not directly stratify participants based on whether they had outdoor or indoor jobs in our analysis.

Nevertheless, our measure of Sunlight Exposure Duration (SED) was designed to capture the behavioral dimension of sunlight affinity by asking participants about the time they spent outdoors (and not under shade) between 9:00 am and 5:00 pm over the past 30 days, including both working and non-working days. This approach aimed to capture each individual's total habitual sunlight exposure, regardless of its source (occupational or recreational).

That said, we acknowledge that occupational sunlight exposure is a significant contributor to total exposure and may have distinct effects. To partially address this, we included Poverty Income Ratio (PIR) as covariates in our multivariate models, which may indirectly capture some socioeconomic aspects related to occupation type. Furthermore, the positive correlation we observed between SPS (psychological preference) and SED (behavioral exposure) suggests that individuals' attitudes align with their behavior, to some extent, irrespective of job type.

We agree with the reviewer that future studies specifically designed to classify occupational sunlight exposure (e.g., using Standard Occupational Classification codes) would provide valuable additional insights. We have added sentences in the Limitations section of the revised manuscript to acknowledge this point. Thank you for this valuable suggestion.

Revised Manuscript with Track Changes (lines 442-445):

“Third, NHANES does not release detailed occupational codes that would allow us to directly classify participants into outdoor versus indoor jobs. Consequently, we could not explore whether the observed associations differ between men with occupational sunlight exposure and those primarily working indoors. Future cycles of NHANES or linked occupational databases could help address this question.”

Responses to Comments of Reviewer #2:

Comment 1: I suspect this is evaluating natural sunlight and not a sun lamp or other artificial sun. You may add a clear statement on this as many who live in northern parts of the country regularly use artificial sun lamps.

Response:

We are grateful to the reviewer for raising this salient issue. The reviewer is correct that our study specifically evaluated exposure to natural sunlight, not artificial light sources such as sun lamps or light therapy devices. Our measure of Sunlight Exposure Duration (SED) was based solely on participants' self-reported time spent outdoors and not under any shade between 9:00 am and 5:00 pm. This was designed to capture habitual exposure to natural daylight.

We acknowledge that the use of artificial bright light therapy is a common and relevant intervention, particularly in regions with limited natural sunlight. The absence of data on artificial light exposure is a limitation of our study, as it could potentially confound the observed associations.

In response to this comment, we have now added a clear statement in the Discussion section of the manuscript. Thank you for this valuable suggestion, which has helped to improve the clarity and scope of our manuscript.

Revised Manuscript with Track Changes (lines 445-449):

“Fourth, our measure of sunlight exposure (SED) captured only exposure to natural sunlight during daytime hours. We did not have data on exposure to artificial light sources, such as bright light therapy lamps or full-spectrum lighting, which are commonly used, particularly in northern latitudes or for treating seasonal affective disorders. The effects of such artificial light sources on depression and sleep disorders may differ from those of natural sunlight and could represent a potential confounding factor not accounted for in our analysis.”

Comment 2: OSA is mentioned in the introduction but no part of the modeling for the logistic regression. Other sleep disorders such as OSA, particularly untreated OSA, restless legs, narcolepsy, etc. would have a large impact on the length and quality of sleep.

Response:

We would like to express our sincere gratitude to the reviewer for this insightful comment. The reviewer's assertion regarding the significance of clinically diagnosed sleep disorders, such as obstructive sleep apnea (OSA), restless legs syndrome, and narcolepsy, as critical confounders is indeed valid. These conditions have the capacity to exert a substantial influence on both sleep duration and quality.

In our analysis, we aimed to adjust for a broad range of health conditions. However, as the reviewer noted, we did not include these specific sleep disorders as covariates in our logistic regression models. This was primarily due to limitations in the NHANES dataset; while it contains data on general "trouble sleeping" and sleep duration, it does not include comprehensive, clinically verified diagnoses for specific sleep disorders like OSA across all survey cycles used in our study.

Despite this limitation, we attempted to indirectly account for the risk of such conditions by adjusting for several known strong correlates and risk factors that are available in NHANES, including:

Body Mass Index (BMI): A major risk factor for OSA [1].

Hypertension: A common comorbidity of OSA [2].

Cardiovascular Disease (CVD): Also linked to OSA [3].

Diabetes: Another condition associated with sleep disorders [4].

While this approach is undoubtedly imperfect, it was our best available method to partially control for the underlying physiological risks associated with sleep disorders like OSA.

We fully agree with the reviewer that the inability to fully adjust for specific sleep disorders is a limitation of our study, as residual confounding may remain. We have now explicitly acknowledged this important limitation in the revised Discussion section. We strongly advocate that future prospective studies incorporate polysomnography or detailed clinical sleep diagnoses to better disentangle these complex relationships. Thank you for highlighting this crucial point, which significantly strengthens the interpretation of our findings.

Revised Manuscript with Track Changes (lines 449-453):

“Fifth, our models did not adjust for specific clinical sleep disorders such as obstructive sleep apnea, restless legs syndrome, or narcolepsy, which are strong determinants of sleep duration and quality. The NHANES database is deficient in systematic data regarding these clinical diagnoses for all participants. Although we adjusted for key risk factors and comorbidities like BMI, HTN, and CVD, which are proxies for conditions like obstructive sleep apnea, residual confounding remains a possibility.”

References

1. Esmaeili N, Gell L, Imler T, Hajipour M, Taranto-Montemurro L, Messineo L, et al. The relationship between obesity and obstructive sleep apnea in four community-based cohorts: an individual participant data meta-analysis of 12,860 adults. eClinicalMedicine. 2025;83: 103221. doi:10.1016/j.eclinm.2025.103221

2. Brown J, Yazdi F, Jodari-Karimi M, Owen JG, Reisin E. Obstructive Sleep Apnea and Hypertension: Updates to a Critical Relationship. Curr Hypertens Rep. 2022;24: 173–184. doi:10.1007/s11906-022-01181-w

3. Nguyen D, Hoang C, Truong T, Nguyen D, Lam HG, Sharma A, et al. Multi-level phenotypic models of cardiovascular disease and obstructive sleep apnea comorbidities: A longitudinal Wisconsin sleep cohort study. PLoS One. 2025;20: e0327977. doi:10.1371/journal.pone.0327977

4. Khalil M, Power N, Graham E, Deschênes SS, Schmitz N. The association between sleep and diabetes outcomes – A systematic review. Diabetes Research and Clinical Practice. 2020;161: 108035. doi:10.1016/j.diabres.2020.108035

Comment 3: Why was the maximum age 59 years? Older individuals experience high rates of depression in this group and may change the data.

Response:

We thank the reviewer for this important question regarding the age range of our study population.

The selection of participants aged 20 to 59 years was not an arbitrary choice by our research team, but rather a direct consequence of the NHANES survey design itself. The core variables defining our exposure of interest, Sunlight Preference Score (SPS) and Sunlight Exposure Duration (SED), were only collected by NHANES from adult participants aged 20 to 59 years. These questions were not administered to participants aged 60 years and older in the survey cycles we utilized (2009-2020). Therefore, to ensur

---

## [Editor Report · Decision Letter 1]

30 Sep 2025

Associations of sunlight affinity with depression and sleep disorders in American males: Evidence from NHANES 2009-2020

PONE-D-25-14823R1

Dear Dr. Yang,

We’re pleased to inform you that your manuscript has been judged scientifically suitable for publication and will be formally accepted for publication once it meets all outstanding technical requirements.

Kind regards,

Mohammad Hossein Ebrahimi

Academic Editor

PLOS ONE
---

## [Editor Report · Acceptance letter]

PONE-D-25-14823R1

PLOS ONE

Dear Dr. Yang,

I'm pleased to inform you that your manuscript has been deemed suitable for publication in PLOS ONE. Congratulations! Your manuscript is now being handed over to our production team.

Kind regards,

on behalf of

Dr. Mohammad Hossein Ebrahimi

Academic Editor

PLOS ONE